# Evidence of Neutrophils and Neutrophil Extracellular Traps in Human NMSC with Regard to Clinical Risk Factors, Ulceration and CD8^+^ T Cell Infiltrate

**DOI:** 10.3390/ijms251910620

**Published:** 2024-10-02

**Authors:** Linda-Maria Hildegard Moeller, Carsten Weishaupt, Fiona Schedel

**Affiliations:** Department of Dermatology, Skin Cancer Center, University Clinic Münster, Von-Esmarch-Str. 58, 48149 Münster, Germany; linda-maria.moeller@ukmuenster.de (L.-M.H.M.); fiona.schedel@ukmuenster.de (F.S.)

**Keywords:** neutrophil extracellular traps, non-melanoma skin cancer, neutrophils, cytotoxic T cells, cutaneous squamous cell carcinoma, basal cell carcinoma, Merkel cell carcinoma

## Abstract

Non-melanoma skin cancers (NMSC), including basal cell carcinoma (BCC), cutaneous squamous cell carcinoma (cSCC), and Merkel cell carcinoma (MCC), are increasingly common and present significant healthcare challenges. Neutrophil extracellular traps (NETs), chromatin fibers expulsed by neutrophil granulocytes, can promote immunotherapy resistance via an impairment of CD8^+^ T cell-mediated cytotoxicity. Here, to identify a potential therapeutic target, we investigate the expulsion of NETs and their relation to CD8^+^ T cell infiltration in NMSC. Immunofluorescence staining for neutrophils (CD15) and NETs (H3cit), as well as immunohistochemistry for cytotoxic T cells (CD8^+^) on human cSCCs (*n* = 24), BCCs (*n* = 17) and MCCs (*n* = 12), revealed a correlation between neutrophil infiltration and ulceration diameter in BCC and MCC, but not in cSCC. In BCC and cSCC, neutrophil infiltration also correlated with the cross-sectional area (CSA). NETs were not associated with established risk factors but with the presence of an ulceration, and, in cSCC, with abscess-like structures. CD8^+^ T cell infiltration was not reduced in tumors that were NET-positive nor in those with a denser neutrophil infiltration. This study is the first to report and characterize NETs in NMSC. Thus, it gives an incentive for further research in this relevant yet understudied topic.

## 1. Introduction

Non-melanoma skin cancers (NMSC) pose an increasing problem to health care systems. They are the most common cancer in white populations and their current incidence rate is anticipated to continue to rise rapidly in the future [1,2,3,4]. The most prevalent forms are basal cell carcinoma (BCC) and cutaneous squamous cell carcinoma (cSCC) whereas Merkel cell carcinoma (MCC) represents a less common, but highly aggressive type of NMSC. While they share UV radiation as the most important risk factor [5,6] they differ in their cell type of origin and their metastatic behavior.

With an estimated metastasis rate of less than 1%, BCC is the least aggressive yet most common form of NMSC [7]. MCC, on the other hand, with 27% of patients already presenting nodal metastases at diagnosis, is the most aggressive type of NMSC [8]. Alongside UV as a cause, the tumor is in approximately 80% of cases associated with polyomavirus [9,10]. In particular, cSCCs have been disproportionately increasing in recent years [11]. Also, cSCC is reported to show recurrent behavior and form metastases in about 4% of the general population [12,13] and in up to 8% of immunosuppressed organ transplant recipients [14]. In the latter patients, 5-year survival rates are only 23% [15]. While surgical excision and radiotherapy represent the first line therapy in local primary NMSC, checkpoint-inhibitors (Anti-PD-1/ Anti-PD-L1) have recently evolved as a promising therapeutic strategy for locally advanced and metastatic cSCC, BCC, and MCC [16,17,18]. Still, only about half of the patients profit from this therapy [19,20,21], indicating a need for further improvement.

Due to sun exposure, NMSC carries a high mutational burden that results in the generation of multiple neoantigens that can be detected by the immune system [22]. In addition, the association of aggressive NMSC with immunosuppression suggests an important role of the immune system in combating these cancers. Tumor-associated neutrophils (TANs) have been described to exert pro- and antitumor effects but are predominantly associated with poorer overall survival in many types of cancer [23]. Moreover, a high neutrophil-to-lymphocyte ratio has been shown to be a negative prognostic marker also for NMSC [24,25,26]. Thus, neutrophils might serve as target cells for NMSC treatment. Although an inhibiting interaction with cytotoxic CD8^+^ T cells has been shown, the exact mechanisms remain unclear [27].

An emerging subject in recent years has been the ability of TANs to expulse “neutrophil extracellular traps” (NETs). These consist of extracellular chromatin fibers decorated with citrullinated histones and a set of neutrophil proteins such as elastase, myeloperoxidase (MPO), and metalloproteinase 9 [28,29]. First described as a mechanism to entrap and kill bacteria, they have recently been detected in various tumor tissues, e.g., malignant melanoma [30], colon cancer [31], lung cancer, breast cancer, and mesothelioma [32] and are increasingly postulated to play a supportive role in tumor initiation, progression, and metastasis [29,33,34].

In a mouse model, NETs have been shown to physically shield the tumor cells from CD8^+^ T cell and natural killer (NK) cell cytotoxicity [35]. Furthermore, they were found to induce an exhausted phenotype of CD8^+^ T cells by incorporating the immunosuppressive ligand PD-L1 [36]. The abrogation of NETs via DNase I could enhance the response to an anti-PD-1 blockade by increasing the CD8^+^ T cell infiltration and cytotoxicity [37].

Hence, as there have not been any studies so far in NMSC, this work aimed at exploring the expulsion of NETs by neutrophils and their relation to cytotoxic CD8^+^ T cells in human NMSC. Due to their high incidences, their ability to metastasize and the availability of approved immunotherapies, we focused on BCC, cSCC, and MCC.

## 2. Results

A total number of 53 primary skin tumors were included in the study. Twenty-four of them were cutaneous SCCs, with a mean invasion depth of 4.85 (SD = 2.49; min = 1.1, max = 10.0) mm, a mean maximal diameter of 13.51 (SD = 7.1, min = 3.75, max = 25.86) mm, and a mean patient age of 77.9 (SD = 55.3, min = 55.2, max = 92.7) years at the time of excision. Eighteen (75%) of them were ulcerated, with a mean ulceration diameter of 9.37 (SD = 10.09, min = 0.25, max = 34.4) mm as measured in H&E-stained slides (Appendix A).

Seventeen BCCs were analyzed, with a mean invasion depth of 2.34 (SD = 1.17, min = 0.7, max = 5.0) mm and a mean patient age of 73.6 (SD = 9.1, min = 47.3, max = 88.4) years at the time of excision. Nine (52.94%) showed ulceration, with a mean diameter of 4.02 (SD = 2.75, min = 1.71, max = 9.14) mm.

The twelve MCCs stemmed from patients with a mean age of 81.2 (SD = 5, min = 74.6, max = 89.3) years and showed a mean maximal diameter of 14.71 (SD = 4.95, min = 9.06, max = 24.54) mm. Three (25%) were ulcerated, with a mean diameter of 15.17 (SD = 9.5, min = 7.32, max = 25.74) mm. Further information on the study population can be found in Appendix A.

### 2.1. Neutrophil Infiltration Differs between Tumor Entities

Since neutrophils have been shown to play a negative prognostic role in a variety of solid cancers, we first analyzed and compared the neutrophil infiltration in cSCC, BCC, and MCC.

With 37.5% (*n* = 9) of the specimens containing a massive (N3) and 50% (*n* = 12) medium (N2) infiltration according to the score suggested by Weide at al. [38] (N0 = no or only sporadic infiltration, N1 = several signals, N2 = medium infiltration, N3 = massive infiltration, please find additional information in materials and methods, Figure 1A), cSCC showed the strongest infiltrate, followed by BCC with no “N3” tumor but in nearly half of the cases an “N2” infiltration (47.1%; *n* = 8). In the MCCs “N3” could only be detected in one tumor and “N2” in three tumors (25%), while in 50% (*n* = 6) a score of “N1” was found. However, there was a certain heterogeneity in infiltration density not only across the entities but also inside one specific tumor type (Figure 1B, C).

### 2.2. Neutrophil Infiltration Correlates with Ulceration and Tumor Size

In order to identify potential factors that can influence the neutrophil infiltration density and thus explain this heterogeneity, we investigated selected histopathological and clinical factors.

In primary melanoma, neutrophil infiltration is associated with the presence of an ulceration [30]. We found this to apply also to BCC and MCC but not to cSCC. The ulceration diameter was measured on H&E-stained slides and revealed a positive correlation with the infiltration of neutrophils in BCCs (Spearman’s *ρ* = 0.7805; two-tailed *p* < 0.001) (Figure 2A,B). The maximum neutrophil score reached in BCCs was “N2” and all of the BCCs “N2” tumors were ulcerated (*n* = 8). Consistent with these findings, all the tumors that showed no or only sporadic neutrophil infiltration (“N0”, *n* = 3) were non-ulcerated.

The same was true for the MCCs: Correlation of ulceration diameter and infiltration revealed a significant association (Spearman’s *ρ* = 0.7621; two-tailed *p* = 0.0045) (Figure 2B) and the only tumor with an “N3” infiltration was indeed ulcerated. All the specimens with no or sporadic (“N0”) neutrophils (“N1”) infiltration throughout the tumor were non-ulcerated.

Interestingly, neutrophil invasion did not correlate with the extent of ulceration in cSCCs (Spearman’s *ρ* = 0.2291; two-tailed *p* = 0.2816) (Figure 2B).

In order to assess whether a higher neutrophil infiltration predicts a poorer clinical outcome, we analyzed whether cSCCs with established clinical risk factors like immunosuppression, invasion depth ≥6 mm, or grading ≥G3 had a higher neutrophil score. Of note, infiltration density did not differ between the specimens with one or more risk factors and those without (*p* = 0.8042) (Appendix A).

Furthermore, the relationship between neutrophil infiltration and tumor size was analyzed.

The sizes of the tumors were assessed by measuring the cross-sectional area (CSA) on the H&E-stained slides (Figure 2A). With a mean CSA of 102.63 (SD = 58.47, min = 22.0, max = 241.67) mm^2^, MCCs showed the largest area, followed by cSCCs with 64.4 (SD = 51.46, min = 3.63, max = 176.63) mm^2^, and BCCs with 4.88 (SD = 5.04, min = 0.49, max = 16.06) mm^2^ in the mean (Figure 2C, Appendix A).

Notably, neutrophil infiltration did show a correlation with tumor size in BCC (Spearman’s *ρ* = 0.6024; two-tailed *p* = 0.0124) and, though less pronounced, in cSCCs (Spearman’s *ρ* = 0.4835; two-tailed *p* = 0.0167). In MCC, no association between tumor size and neutrophil infiltration could be seen (Spearman’s *ρ* = −0.3506; two-tailed *p* = 0.2638) (Figure 2C).

Another difference between the tumors emerged with regard to the neutrophil distribution: While in BCCs and MCCs the neutrophils were evenly distributed outside the ulcerated area, neutrophils in cSCCs showed a “swarming” behavior. Here, large numbers of neutrophils gather in a swarm-like pattern in the tumor stroma, without the presence of necrosis. Towards the middle of the accumulation, these neutrophils disintegrate, sometimes forming abscess-like structures. (Figure 1C and Appendix A).

This phenomenon was observed in 33.3% (*n* = 8) of the cSCC specimens. Six of them were well differentiated (G1–G2), while the other two were poorly differentiated (G4). The tumors with neutrophil swarming/abscesses showed a greater size than the rest, with a mean CSA of 108.25 mm^2^ vs. 39.91 mm^2^ (two-tailed *p* < 0.001).

### 2.3. NETs Do Not Correlate with Established Risk Factors but with Ulceration

Next, it was investigated whether NETs are formed by neutrophils in the tumors.

Analogous to the neutrophil infiltration, tumors were classified as “NET0” (no NETs), “NET1” (several NETs found), “NET2” (medium NET detection), and “NET3” (massive NET detection) (Figure 3A). While neutrophils could be detected in all of the cSCCs (*n* = 24) and the great majority of BCCs (*n* = 14, 82.35%) and MCCs (*n* = 10, 83.33%), NETs were found in 79.2% (*n* = 19) of the cSCCs, in 58.8% (*n* = 10) of the BCCs, and in 33.3% (*n* = 4) of the MCCs.

The score “NET3” occurred only in three cSCCs and one MCC. In line with the neutrophil infiltration, the MCCs were the tumors with the least NET infiltration, with 66,7% showing no NETs at all (Figure 3B).

In order to explore possible correlations and influencing factors, a more precise analysis of clinical and histological risk factors was conducted. As observed for the neutrophil infiltration, a clear association between the ulceration diameter and the detection of NETs could be seen across all tumor entities (Figure 3C).

The ulceration diameter was significantly bigger in tumors that showed an infiltration with NETs than those that did not (*p* = 0.0091 for cSCC, *p* = < 0.001 for BCC and *p* = 0.0182 for MCC (Figure 3C). Moreover, Spearman’s correlation revealed a positive directed relationship between ulceration diameter and the given NET score (Spearman’s *ρ* = 0.6851, two-tailed *p* < 0.001 for cSCC; Spearman’s *ρ* = 0.8248, two-tailed *p* < 0.001 for BCC; Spearman’s *ρ* = 0.8431, two-tailed *p* = 0.0061 for MCC) (Appendix A). One non-ulcerated BCC and one non-ulcerated MCC contained neutrophils extruding NETs in proximity to a vessel in the subcutaneous fat tissue and underneath the epidermis, respectively.

Without exception, all ulcerated BCCs and MCCs showed NETs close to the ulceration zone and, if present, in the scab (Figure 3B,D). In cSCCs, two ulcerated tumors showed no NET infiltration, and several non-ulcerated tumors showed NET infiltration (*n* = 3). Interestingly, in all these non-ulcerated cSCCs NETs could be detected in the “neutrophil swarms” or abscess-like structures (Appendix A).

Similar to the infiltration with neutrophils, the clinically used risk factors of cSCCs, i.e., invasion depth ≥ 6 mm, Ggrading ≥ 3, and immunosuppression, were not associated with a higher NET-Score (*p* = 0.3176) (Appendix A).

Taken together, NETs could particularly be seen in ulcerated areas. Supporting this finding, we saw a strong association of NETs with an ulceration present in a tumor, especially in BCCs and MCCs. In cSCCs, however, they were not that strictly confined to ulcerated areas and also got expulsed by neutrophils more distant to the surface showing a swarming behavior.

### 2.4. NETs in cSCC Metastases

In this study, we primarily focused on the primary lesions of NMSC, as these represent the origin of metastasis. However, the immunological processes within metastases are of great importance for the clinical course and the success of immunotherapies. Since the formation of NETs in NMSC metastases has not yet been investigated, we conducted an exploratory staining of NMSC metastases.

For the cSCC, six cutaneous metastases and two lymph node metastases were evaluated. They all showed neutrophil infiltration of varying degrees (Appendix A). NETs were detected in ulcerated and necrotic cutaneous metastases, but not in non-necrotic metastases with an intact epidermis (Figure 4A). Interestingly, a lymph node with necrosis did not show any NETs, although highly infiltrated by neutrophils (Figure 4B).

For MCCs, five metastases were stained: Two were lymph nodes, one originated from a cutaneous metastasis, and there was one each from the salivary and adrenal glands. They showed a sporadic to mild infiltration with neutrophils but, in spite of the presence of necrotic areas in the lymph nodes and the adrenal gland metastasis, no NETs (Appendix A).

Thus, we can present the first evidence of NETs in metastases of cSCCs but not in MCC metastases.

### 2.5. NET-Infiltrated Tumors Do Not Show Reduced CD8^+^ T Cell Infiltration

In order to assess whether the expulsion of NETs can influence the immune environment by shielding the tumor from infiltrating, anti-tumor-acting cells, we investigated the infiltration with cytotoxic T-cells. CD8^+^ T cells were counted inside the tumor and in the peritumoral area and the quotient of their densities was calculated (Figure 5A).

Due to staining heterogeneity and consequently imprecise cell counting, two BCCs, four cSCCs, and one MCC had to be excluded from the analysis. Thus, 15 BCCs, 11 MCCs, and 20 cSCCs were analyzed in total.

All entities showed more cytotoxic T cells in the peritumoral area than inside the tumor. For BCC and cSCC, the difference was significant (both *p* < 0.0001), while the MCCs, due to two highly infiltrated outliers, showed only a trend (p = 0.6895) (Figure 5B).

Statistical analyses revealed no difference in the density quotient between NET-positive and NET-negative tumors (*p* = 0.5536 for cSCC, *p* = 0.5358 for BCC, *p* = 0.7758 for MCC). Furthermore, there was no correlation between the neutrophil score and the density quotient of CD8^+^ T cells (Spearman’s *ρ* = 0.102, two-tailed *p* = 0.6690 for cSCC; Spearman’s *ρ* = −0.2243, two-tailed *p* = 0.4218 for BCC; Spearman’s *ρ* = 0.2849, two-tailed *p* = 0.3957 for MCC) (Figure 5C, D). Although neutrophils were more present in ulcerated specimens (Figure 2B), no difference in CD8^+^ T cell infiltration was found between specimens with and without ulceration (Appendix A).

Interestingly, no or only sporadic CD8^+^ T cells could be detected in neutrophil-swarming areas and abscesses (Appendix A).

## 3. Discussion

Neutrophils have been shown to be able to modulate the tumor immune environment by different means, often predicting poor survival [23]. One mechanism could be the extrusion of NETs, which in most cases has been found to play a negative prognostic role in multiple steps of cancer progression [33,34]. Specifically, within blood circulation, they can promote metastasis by capturing disseminated tumor cells and facilitating their extravasation [33]. Furthermore, NETs skew the tumor microenvironment towards an immunosuppressive phenotype [33], e.g., by exhausting CD8^+^ cytotoxic T cells, the main effector cells of a PD-1/PD-L1 or CTLA4-directed immune checkpoint blockade [27]. Confirming these findings, abrogating NETs has indeed been shown to improve the efficacy of this therapy in a mouse model [37].

In recent years, research has brought forth several components for practical inhibition of NET formation and their degradation [39]. The inhibition of PAD4, a crucial enzyme for NET formation [40,41], has been shown to delay disease progression in mouse multiple myeloma [42]. Moreover, studies on mouse models of lung and colorectal cancer have revealed decreased metastases under NET inhibition via PAD4 or NE inhibition [43,44]. The degradation of NETs via DNAse I has also been shown effective in improving immunotherapy response in a mouse model of colorectal cancer [37]. Notably, systemic treatment with DNAse I has already been performed in patients and described as safe and well-tolerated [45].

Due to the increasing incidence of NMSCs and their ability to form metastases, especially in immunosuppressed organ transplant recipients, it is of crucial importance to identify additional therapeutic targets. We therefore aimed to characterize the presence of neutrophils and NETs in cSCC, BCC, and MCC and their relation to infiltration with CD8^+^ T cells.

We found neutrophils to be detectable to various extents in the examined tumor entities, with cSCC being by far the most infiltrated. The reason behind this phenomenon remains unclear: IL-8, the most important chemoattractant cytokine for neutrophils, has been shown to be elevated in both SCC and BCC, without a significant difference between these entities [46,47]. We showed that the infiltration of cSCCs with neutrophils does not depend on the presence of an ulceration, thus, additional factors must be at play. An explanation could be the generally larger size of cSCCs, as there is a known positive correlation between size and neutrophil infiltration [48]. Although research comparing the immune infiltrate in different types of NMSC has been scarce, an increased infiltration also with CD68^+^ macrophages and CD8^+^ T cells in cSCC compared to BCC suggests a generally more immunogenic environment [49].

Of note, the present study is the first one to report NETs in NMSC, the most common cancer in fair-skinned populations, laying the foundation for further research in this promising area for the improvement of immunotherapy. In 79.2% of cSCCs, 58.8% of BCCs, and 33.3% of MCCs, NETs were present. In high-risk and metastatic cSCC and MCC, NETs could therefore indeed represent a potential therapeutic target.

Here, NETs are shown to be associated with the presence of an ulceration across all entities. This falls in line with former publications about melanoma primaries, where NETs were found only in ulcerated specimens [30]. The mechanism, however, remains not fully understood. One potential explanation could be the initial damage of the epithelium due to the tumor cells’ inability to maintain a physiological barrier and subsequent activation by neutrophils by bacterial antigens from the surface. Once extruded, NETs and their attached proteases as neutrophil elastase can further promote the damage of the extracellular matrix leading to or contributing to the formation of an ulceration. Indeed, they have been shown to be able to directly kill endothelial and epithelial cells [50,51]. Furthermore, NETs can delay the wound closure in wound healing. In mice that were deficient in PAD4, a crucial enzyme for NET formation [40,41], wounds healed significantly faster through an accelerated re-epithelialization [52].

Taken together, these findings give ideas to the underlying mechanisms why NETs were predominantly present in ulcerated areas. However, the exact mechanism and the order in which NETs and ulceration occur, whether one precedes the other or they develop simultaneously, has yet to be investigated.

Furthermore, it remains unsolved why NETs can also be detected in the absence of ulceration: In 50% (*n* = 3 out of *n* = 6) of the non-ulcerated cSCCs NETs were found, all localized in neutrophils “swarming” areas and abscesses in keratin pearls without an associated microscopically distinguishable necrosis. Neutrophil swarming has been described in various tissues but requires some first triggering factor like a pathogen or sterile injury [53]. In oral SCC, keratin pearls have been proposed to form due to the rupture of intraepithelially entrapped blood vessels (IEBVs) resulting in erythrocyte-induced oxidative stress on the cancer cells that leads to increased keratinization [54]. The degradation of the keratin pearls was shown to be reciprocally regulated by neutrophils and macrophages [55]. Although this mechanism has not been proven in cutaneous SCC, keratin pearls constitute one of the histological characteristics also in these tumors, usually seen as a sign of good differentiation [56,57]. In our study, two out of three non-ulcerated cSCC with NETs were well-differentiated with a grading of G1. It appears reasonable that neutrophils recruited to sites of ruptured IEBVs get activated, exhibit a swarming behavior and extrude NETs which, in turn, could increase the oxidative stress on the cancer cells. Indeed, NETs have been shown to entrap and fragment red blood cells in mouse models and in vitro [58,59,60].

Tumor cells themselves, on the other hand, may induce NET formation by excretion of cytokines. In vitro, NET formation in neutrophils was induced by interleukin-8 (IL-8) in conditioned media of a human melanoma cell line and an anaplastic thyroid cancer cell line [61,62] and with the plasma of patients with colorectal cancer [63]. Furthermore, granulocyte-colony-stimulating factor (G-CSF) expressed by tumor cells has been proposed to predispose neutrophils to release NETs [64]. Hirai et al. found elevated G-CSF expression in human cSCCs [65]. In contrast, Lee et al. suggest that non-invasive cell lines barely expressed G-SCF [66]. Thus, G-CSF could facilitate the NET-extrusion in cSCC but not in the, usually non-invasive, BCC.

In order to learn more about the effect of NETs in NMSCs on the T cell immune response we explored whether NETs can lead to immune exclusion of cytotoxic T cells in NMSCs. In line with Frohwitter et al., significantly more CD8^+^ positive cells could be found peritumorally than inside the tumor in BCCs and cSCCs [49]. Of note, in swarming areas with higher neutrophil densities and NETs no CD8^+^ T cells could be distinguished. However, on a whole tumor level, no significant difference in CD8^+^ lymphocyte infiltration dependent on the quantity of neutrophil or NET infiltration was found.

This diverges from recent studies in other tumor tissues that report a negative correlation between NET infiltrate and CD8^+^ T cells [32,67]. The research on neutrophils, however, is ambiguous: Studies report a colocalization [68] and even costimulation of CD8^+^ T cells and neutrophils [69]. Given the complex nature of the tumor immune microenvironment, additional factors such as an increase in neoantigens due to a high mutational burden in skin cancers [22] and a distinct microbiome [70] are likely to contribute as well and help explain these diverging findings.

A possibly dampening effect of NETs on T cells by functional exhaustion rather than spatial exclusion, e.g., through an increased NET-associated PD-L1 expression, as previously described by Khou et al. [27] and Kaltenmeyer et al. [36], cannot be excluded and requires further investigation.

Importantly, neutrophil granulocytes display a fairly heterogeneous and plastic population of cells that can be shaped towards distinct phenotypes like the more anti-tumorigenic (N1) and the more pro-tumorigenic (N2) phenotype [71]. However, due to the use of the general granulocyte marker CD15, a carbohydrate on the surface that allows tethering on endothelial and dendritic cells, this study cannot make conclusions about the exact neutrophil phenotypes in NMSC. Further investigations, e.g., via gene expression analysis, would be needed to compare their polarization and activation status, e.g., in different regions of the tumors. Moreover, a more precise quantification of the neutrophil and NET infiltrate by implementing a reliable (semi-) automatic workflow would be desirable.

Other limitations of the study are the retrospective design and the small number of MCC specimens and NMSC metastases. Due to limited follow-up information on patients with cSCCs and BCCs, no prognosis analysis could be performed. It would be interesting to see whether our findings can be confirmed in a study with a larger cohort and correlated with a complete clinical follow-up.

Because of their clinical relevance, we focused on the NMSC entities cSCC, BCC, and MCC. Other, less common types of NMSC, however, were not evaluated and the proof and role of neutrophils and NETs in these tumors might differ from the findings of this study.

Taken together, the present study is the first to show an extrusion of NETs in human NMSC primary tumors and cSCC metastases. Interestingly, we found an association of NETs with an ulceration, but a diverging pattern and amount of neutrophil infiltration in BCC, MCC, and cSCC, suggesting different immune environments.

It deserves further investigation to discover whether a combination of NET inhibitors with established immunotherapy or radiotherapy can improve therapy response and reduce metastases also in NMSC. Even more, as NETs also play a physiological role in fighting infectious diseases, the effect of NET inhibition on the immune defense in immunocompromised patients, the most vulnerable group for high-risk skin tumors, needs to be evaluated with special attention.

Considering this, along with the increasing incidence of NMSC and consequently rising numbers of patients in advanced stages, the investigation of neutrophils and NETs in the complex immune response to NMSC gains high importance.

## 4. Materials and Methods

### 4.1. Patient Samples

The patient database of the Department of Dermatology, Münster University Hospital (Münster, Germany) was screened for patients with MCC, cSCC, and BCC and available samples from 2010 to 2022 meeting the following criteria: Complete excision, complete medical history regarding pharmacological treatment at the time of excision as well as secondary diseases, and complete pathological assessment (written report, localization, grading, minimum invasion depth, and staging).

For the selected specimens, formalin-fixed paraffin-embedded (FFPE) tissue blocks along with their corresponding H&E staining were obtained.

### 4.2. Immunofluorescence Staining

FFPE tissue blocks were cut into 3 µm sections and dried on a heating plate for >4 h. Following automated dewaxing, antigen retrieval was performed in citrate buffer pH = 6 ± 0.2 (DCS #CL009C500, Hamburg, Germany) for 20 min in a steam stove followed by 20 min at room temperature (RT). Blocking was performed with 1% bovine serum albumin (BSA) and 10% normal goat serum in PBS for 30 min. Subsequently, sections were incubated at 4 °C overnight with the primary antibodies H3Cit, an established marker for NETs in all stages of NET formation (Abcam #5103 Cambridge, UK, 1:50) and CD15, a granulocyte marker that has been proven to be a valid marker for neutrophils by previous co-staining with MPO and neutrophil elastase in our lab [30] (Biolegend #301902, San Diego, CA, USA, 1:50). This was followed by a 45 min incubation at RT with the secondary antibodies AF568 labeled goat-anti-rabbit (Thermo Fisher Scientific #A11036, Waltham, MA, USA, 1:500) and AF488 labeled goat-anti-mouse (Abcam #150121, Cambridge, UK, 1:300). DAPI (Roche #10236276001, Basel, Switzerland, 1:1800) was added for counterstaining. Desiccation of the specimens was attentively avoided by handling no more than ten slides at once.

Sections were mounted using Fluorescence Mounting Medium (Agilent #S302380-2, Santa Clara, CA, USA) and Carl Roth Coverslips (thickness 1.5, Carl Roth, #KCY5.1, Karlsruhe, Germany). In all experiments, a NET-positive primary melanoma served as a positive control. Additionally, an adjacent slice of every specimen was stained with isotype antibodies (rabbit IgG: Biolegend #910801, San Diego, CA, USA, 1:50 and mouse IgM: Biolegend #401601, San Diego, CA, USA, 1:50) as negative controls.

Between steps, washing was performed in freshly prepared washing buffer (DCS #WL583C2500, Hamburg, Germany) diluted as recommended in distilled water. Probes were stored at 4 °C and assessed in a timely manner.

### 4.3. Immunohistochemistry

Staining for CD8 on T lymphocytes (Agilent #M7103, Santa Clara, CA, USA, 1:100) was performed with the Autostainer Lab vision 480S-2D (Thermo Fisher Scientific #15869452, Waltham, MA, USA) on serial sections of the specimen.

### 4.4. Image Acquisition and Analysis

The assessment of specimen and fluorescence image acquisition was conducted on the Olympus BX63 fluorescence microscope (Olympus, Hamburg, Germany). H&E and immunohistochemical stained slides were scanned with the Epredia™ Lab Vision™ Autostainer 480S-2D (Epredia Thermo Fisher Scientific #A80500027, Waltham, MA, USA).

A scoring system with categories from 0 to 3 for the density of neutrophil infiltration and a separate one for the NET density detected in a tumor was used as previously described [38]. According to the signal detected in the green channel (CD15, density of neutrophil infiltration) or the red channel (H3cit, density of NETs) respectively, the tumor was labeled as “N0” or “NET0” (no or only sporadic infiltration), “N1” or “NET1” (several signals), “N2” or “NET2” (moderate infiltration) or “N3” or “NET3” (massive infiltration) (Figure 1B). In contrast to automatic quantification by thresholding and counting positive pixels, this allowed the assessment of the specimen as a whole while adjusting for different tissue properties and excluding artifacts.

In order to ensure an objective assessment, each sample was evaluated at least twice in a blinded fashion. In case of differing results, the specimen was assessed a third time in comparison with samples already evaluated that had been assigned the two scores in debate.

Image analysis was conducted using the Fiji Software 2.9.0/1.53t and the respective plugins [72] as well as the open-source software QuPath 0.4.3 [73].

The ulceration of the tumor was assessed on the H&E staining. If a thinning of the epidermis with interspersing immune cells and/or a discontinuity of the epidermis was observed, the specimen was labeled as “ulcerated” and the length of the ulceration was documented. If the epidermis could be distinguished throughout the whole surface, it was labeled as “non-ulcerated”.

The cross-sectional area (CSA) was measured by annotating all the areas in the section where tumor cells could be detected.

Assessment of cytotoxic T-cell infiltration was performed with the positive cell detection command in QuPath. Positive cells inside the tumor and in the peritumoral area were counted separately. Intratumoral cytotoxic cells were defined as CD8^+^ cells in direct contact with tumor cells. For the peritumoral area, a 300 µm rim around the tumor, as annotated in the CSA, was defined, since in that range, the densest infiltrate could be found (Figure 5A). Ulceration, epidermis, sebaceous glands, cystic structures, as well as hemorrhages were excluded.

The density quotient (DQ) was calculated as follows:(1)$$\mathrm{DQ}=\frac{\mathrm{intratumoral}{\mathrm{CD}8}^{+}\mathrm{Tcells}/{\mathrm{mm}}^{2}}{\mathrm{peritumoral}{\mathrm{CD}8}^{+}\mathrm{Tcells}/{\mathrm{mm}}^{2}}\times 100$$

### 4.5. Statistical Analysis

Statistical analysis was done with IBM SPSS Statistics (Version 29.0.0.0, IBM Corporation, Armonk, NY, USA) and GraphPad Prism (Version 9.4.0, San Diego, CA, USA). Testing for normal distribution was performed by Shapiro-Wilk analysis. In order to assess the correlation between two non-normally distributed variables, Spearman’s *ρ* was used. Differences between the two groups were analyzed with the unpaired t-test for normally distributed variables and the Mann-Whitney-U-Test for non-normally distributed variables, respectively. Values of *p* < 0.05 were considered statistically significant.

## Figures and Tables

**Figure 1 ijms-25-10620-f001:**
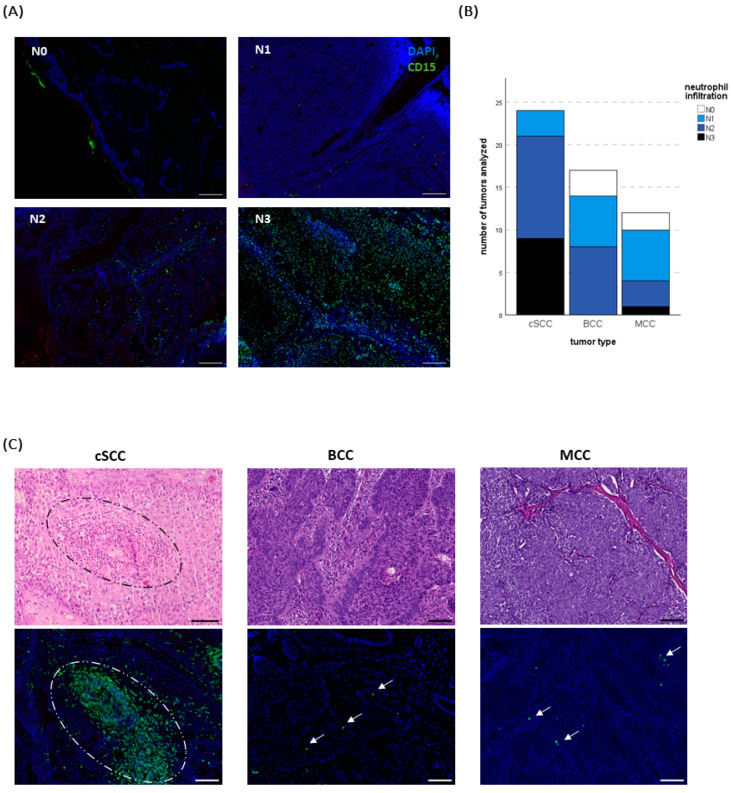
(**A**) Exemplary pictures of different neutrophil scores (N0–N3) assessed by immunofluorescence (IF) staining. CD15 (green) was used as a granulocyte marker, and DAPI (blue) served as counterstaining. Scale bars 200 µm. (**B**) Overview of the analyzed tumor specimens and the detection of neutrophils. (**C**) Representative H&E as well as respective IF images of a cutaneous squamous cell carcinoma (cSCC), a basal cell carcinoma (BCC), and a Merkel cell carcinoma (MCC) (single neutrophils indicated by arrows). Several of the cSCCs showed a high infiltration, in part with a swarming behavior (dotted circle) which was exclusively present in cSCCs. Scale bars 100 µm.

**Figure 2 ijms-25-10620-f002:**
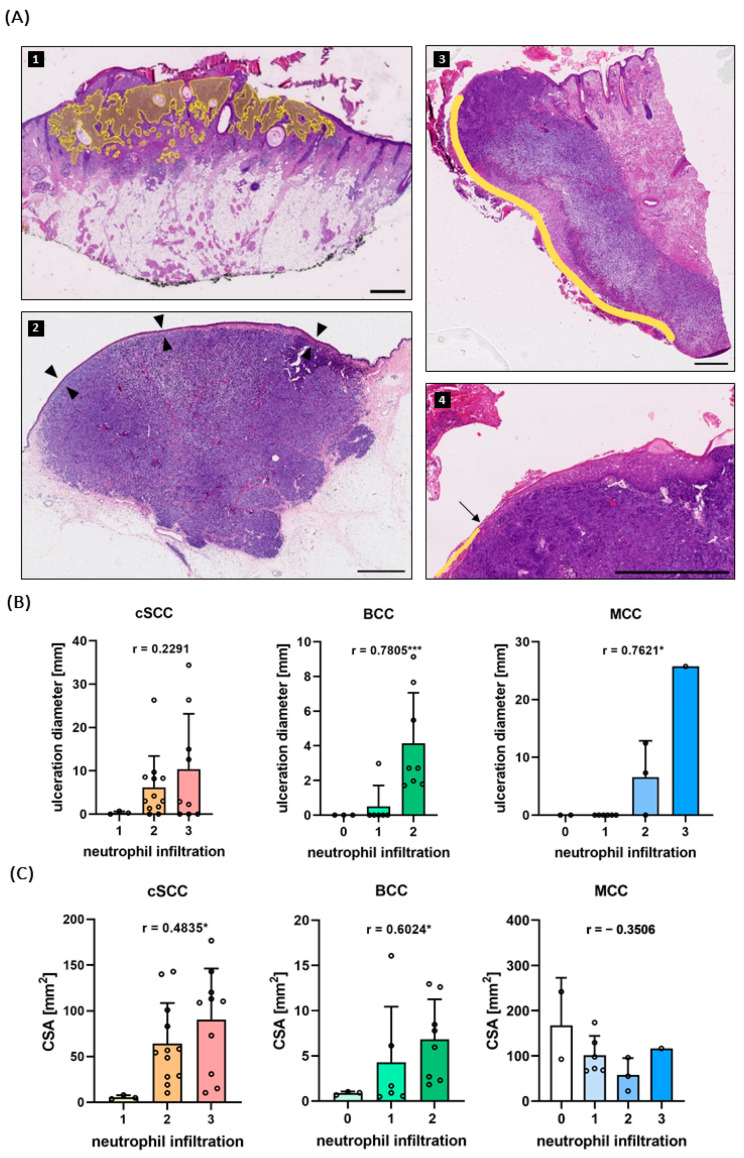
(**A**) (1) Representative H&E staining of a BCC. Tumors were annotated manually on the scanned slides and the cross-sectional area (CSA) was calculated with the open-source software QuPath 0.4.3. (2) A non-ulcerated MCC. If an intact epidermis (arrowheads) could be distinguished throughout the whole sample, the specimen was labeled as non-ulcerated. (3) and (4) Overview and close-up of an ulcerated cSCC. Ulceration (yellow line) was defined as a discontinuity of the epidermis (start indicated by arrow). Scale bar 1000 µm. (**B**) In BCC and MCC the neutrophil score given in the immunofluorescence staining correlated significantly with the ulceration diameter. In cSCC, no correlation was found. (**C**) In cSCC and BCC, but not in MCC, the neutrophil infiltration correlated significantly with the CSA. Bars indicate mean + standard deviation (SD). r = Spearman’s *ρ*. * *p* < 0.05, *** *p* < 0.001.

**Figure 3 ijms-25-10620-f003:**
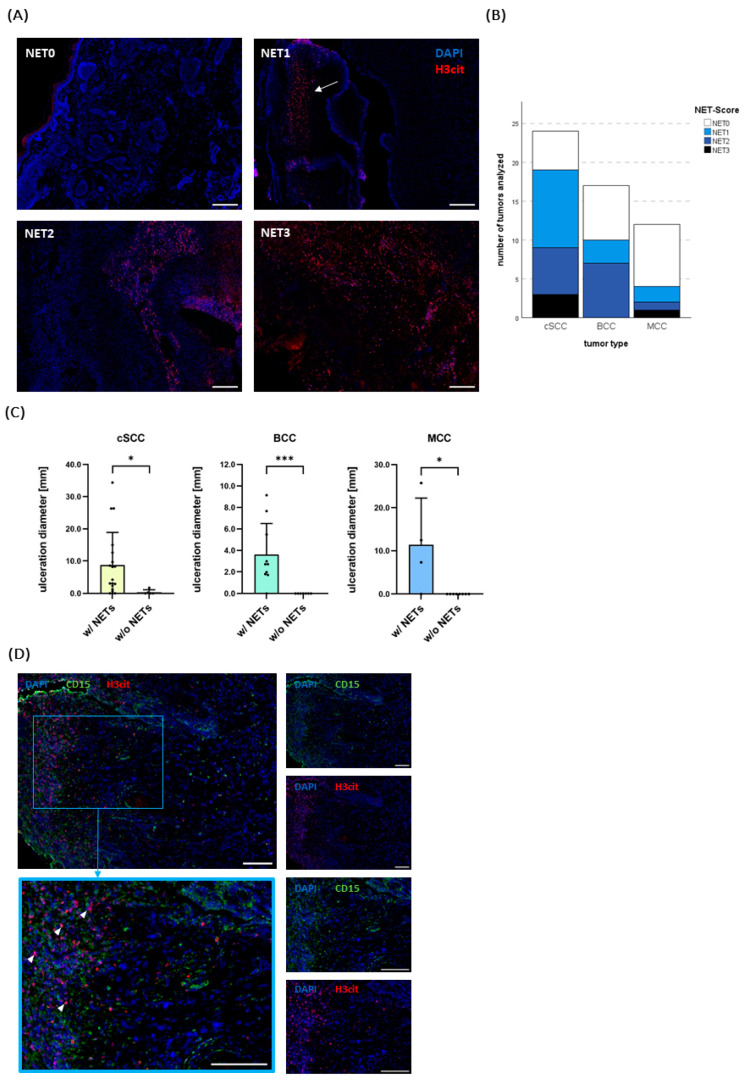
(**A**) Exemplary pictures of different neutrophil extracellular traps (NET) scores (NET0–NET3) assessed by immunofluorescence staining. H3cit (red) indicates NET formation. DAPI (blue) served as counterstaining. NET0: A non-ulcerated BCC with no NETs. NET1: An ulcerated BCC with some infiltration by NETs. In BCCs, they typically could be distinguished in the scab (arrow) and in the ulceration zone. NET2: An ulcerated cSCC with medium NET infiltration. NET3: An ulcerated cSCC with extensive NET infiltration. Scale bar 200 µm. (**B**) Overview of the analyzed tumor specimens and the detection of NETs. (**C**) The presence of NETs was associated with a significantly larger ulceration diameter across all entities. Bars indicate mean + SD. (**D**) Representative immunofluorescence images of a cSCC with a neutrophil score of N3 and a NET score of NET3. CD15 (green) was used to stain granulocytes. Small images show CD15 (green) and H3cit (red) channels separately. Towards the ulceration, disintegrating granulocytes can be distinguished that, in part, expulse NETs (examples shown by arrowheads). Scale bars 100 µm. * *p* < 0.05, *** *p* < 0.001.

**Figure 4 ijms-25-10620-f004:**
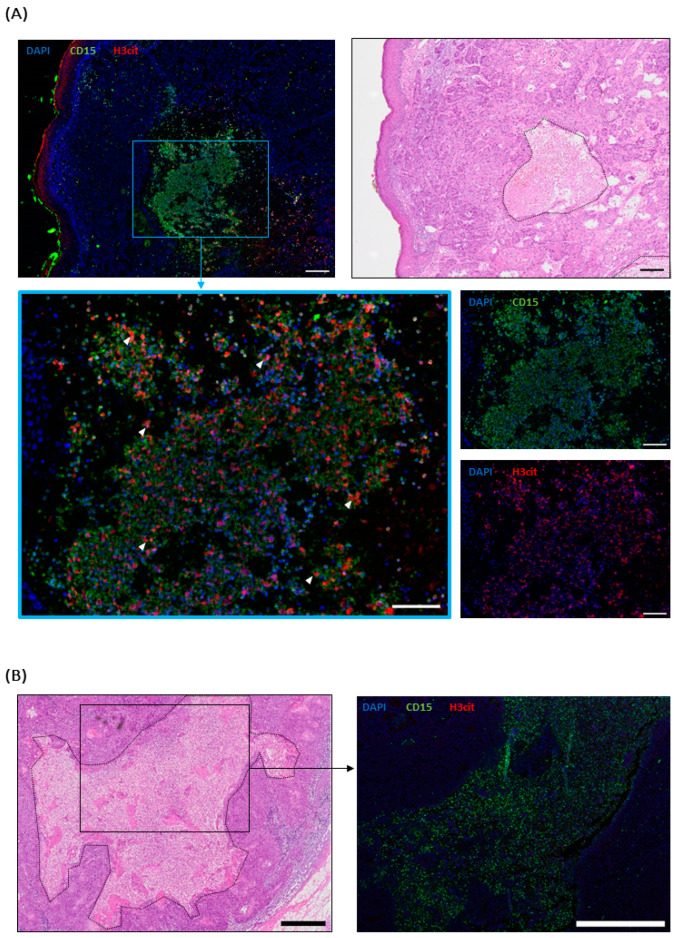
H&E and the respective immunofluorescence (IF) images of exemplary stained cSCC and MCC metastases. CD15 (green) was used as a granulocyte marker, H3cit (red) visualized NETs, and DAPI (blue) served as counterstaining. Necrotic areas are surrounded by dotted lines. (**A**) A necrotic cutaneous cSCC metastasis. Scale bar 200 µm. Enlarged section: Neutrophils in the necrotic area expulse NETs (examples indicated by arrowheads). Scale bar 100 µm. (**B**) A necrotic lymph node, highly infiltrated by neutrophils (green), but without NETs. Scale bar 1 mm.

**Figure 5 ijms-25-10620-f005:**
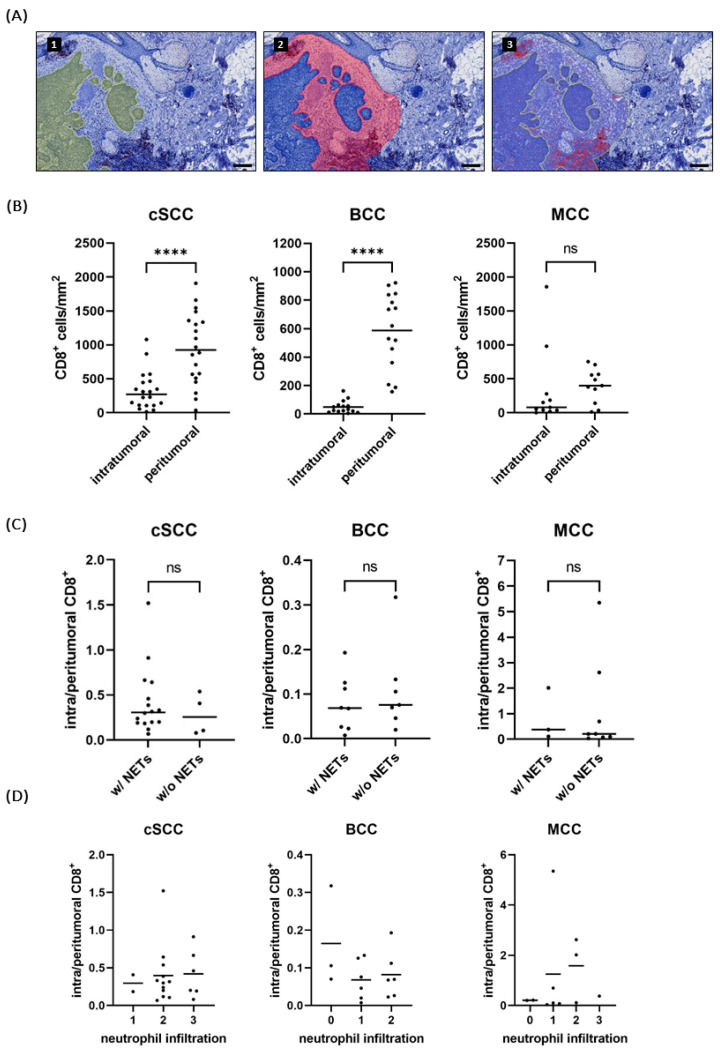
Intra- and peritumoral CD8^+^ T cells were counted via immunohistochemistry and the open-source Software QuPath 0.4.3. (**A**) The tumor (1, green area) was annotated manually and a 300 µm rim (2, red area) was defined as the peritumoral region. Positive cells (3, red dots) were counted with the Positive Cell Detection command. The CD8^+^ T-cell density quotient was calculated by dividing the intratumoral CD8^+^ T cell density [cells/mm^2^] by the peritumoral CD8^+^ T cell density [cells/mm^2^]. Scale bar 200 µm. (**B**) In BCC and cSCC, significantly more CD8^+^ T cells were detected in the peritumoral area than inside the tumor. (**C**,**D**) No difference in the CD8^+^ T cell density quotient was seen in specimens that contained NETs and those that did not, nor in specimens with different neutrophil infiltration densities. **** indicates *p* < 0.0001, ns = not significant.

## Data Availability

The data presented in this article can be found in the main text, figures and the Appendix A. For additional information and raw data the reader is encouraged to contact the corresponding authors. Patient data, however, cannot be handed over.

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
