# Peer review of "Evidence of Neutrophils and Neutrophil Extracellular Traps in Human NMSC with Regard to Clinical Risk Factors, Ulceration and CD8+ T Cell Infiltrate"

_ijms, 2024, doi:10.3390/ijms251910620_

Round 1
Reviewer 1 Report
Comments and Suggestions for Authors
The work addresses an important topic, the need to include neutrophils in the diagnosis of cancer. The manuscript is well written, and correctly refers to the results, although I would point out the limitations of the results.
My question focuses on the choice of CD15 for differentiating neutrophils, it is not an ideal marker. In addition, the CitH3 indicator is recognized as an early stage of neutrophils, but not a direct one, it is worth noting.
The images are very small and illegible, it is difficult to localize neutrophils and NETs.
There are publications available indicating the localization of NETs in human tumor tissue, no such items are cited.
Author Response
Dear editor, dear reviewers, please find our point-by-point response in the attachment.

Reviewer 2 Report
Comments and Suggestions for Authors
The study on neutrophil extracellular traps (NETs) in non-melanoma skin cancer (NMSC) offers significant insights into a relatively unexplored area of cancer immunology. The correlation of NETs with ulceration and the interaction with CD8+ T cells presents an interesting angle that could lead to novel therapeutic strategies, particularly in immunotherapy. The article demonstrates clear expertise in immunofluorescence staining, and the results are well presented. However, several sections require a deeper exploration of the implications of the findings, particularly regarding how NETs can modulate the immune environment. The manuscript would benefit from an expansion of the discussion on clinical relevance and further contextualization of the results within the broader literature.
The authors should elaborate on how the findings of NET formation might lead to practical clinical applications, particularly in improving immunotherapy outcomes or influencing surgical/radiotherapy decision-making in NMSC. This could involve referencing recent advances in targeting NETs for other cancers and applying them to NMSC.
The rationale behind selecting these specific tumour types should be more clearly outlined, especially given the heterogeneity in NET formation across different cancers. Addressing the potential limitations of focusing on these types could improve the clarity and generalizability of the study.
The authors should compare their methodology with other studies that have quantified NETs, discussing the advantages and limitations of the chosen scoring system. Including alternative approaches or justifying the exclusion of other methods would provide more depth to the methods section.
A discussion of tissue handling protocols and their impact on NET preservation is crucial to ensure the reproducibility and reliability of the results. This section should detail specific measures taken to mitigate any potential issues related to sample processing.
The authors should provide a deeper explanation of why NETs correlate with ulceration in BCC and MCC but not cSCC. Exploring potential biological mechanisms that differentiate these tumour types, such as differences in immune microenvironments, would enhance the interpretability of the results.
The lack of a relationship between NETs and CD8+ T cell infiltration contradicts prior findings in other cancers. The authors should explore this discrepancy in more detail, potentially discussing whether the tumour microenvironment in NMSC differs in a way that prevents this interaction or whether other immunosuppressive mechanisms are at play.
The discussion would benefit from a more extensive comparison of NET-CD8+ T cell interactions in NMSC versus other solid tumours. This should include a critical analysis of why the expected immune exclusion did not occur in this study and whether the results align with or diverge from broader cancer immunology literature.
Expanding the discussion to include the implications of NET formation in preventing metastasis would provide a clearer link between the basic science findings and their potential clinical applications. This should include an analysis of how targeting NETs might reduce metastatic risk, especially in immunosuppressed patients who are more prone to aggressive disease.
Each of the questions above should be addressed in detail, and responses should be included in both the manuscript (with changes highlighted in yellow) and in the letter responding to reviewer queries. These changes are critical for the manuscript to be accepted, as they address gaps in methodology explanation, result interpretation, and clinical relevance that are central to the study's impact.
Author Response

(The authors gave the same response as above.)

Round 2
Reviewer 2 Report
Comments and Suggestions for Authors
Thank you for your responses which I fine satisfactory